# Structural Analysis of Composite Conical Convex-Concave Plate (CCCP) Using VAM-Based Equivalent Model

**DOI:** 10.3390/ma14082012

**Published:** 2021-04-16

**Authors:** Qingshan Yi, Yifeng Zhong, Zheng Shi

**Affiliations:** 1School of Civil Engineering, Chongqing University, Chongqing 400045, China; 20191602063t@cqu.edu.cn (Q.Y.); 20191601513@cqu.edu.cn (Z.S.); 2Key Laboratory of New Technology for Construction of Cities in Mountain Area, Chongqing University, Chongqing 400045, China

**Keywords:** composite conical convex-concave plate, variational asymptotic method, finite element analysis, buckling analysis, natural frequency

## Abstract

Compared with the ordinary foundation plate, the composite conical convex-concave plate (CCCP) has obvious anisotropic characteristics, and there is less research on the relationship between its mechanical properties and structural parameters. In this article, a numerical model for the equivalent stiffness of a typical unit cell with conical convex is established by using the variational asymptotic method. Then, the 3D finite element model (3D-FEM) of CCCP is transformed into 2D equivalent plate model (2D-EPM) with the effective plate properties obtained from the constitutive analysis of unit cell. The accuracy of 2D-EPM is verified by comparing with the displacement, natural frequencies, and buckling results from 3D-FEM under different boundary conditions. Then, the influence of geometric parameters and layup configurations on the effective performances of CCCP are investigated. Finally, the buckling loads and natural frequencies of bidirectional CCCP are compared with those of CCCP by using the present model. The present model is particularly useful in the early design stage of CCCP where many design trade-offs need to be made over a vast design space in terms of material selection, ply angles, and geometric parameters.

## 1. Introduction

With the development of modern science and technology, thin plate theory cannot meet the requirement of industrial production. The main disadvantage is that the increase in stiffness can only be achieved by changing the thickness of thin plate, but the thin plate theory is no longer applicable when the thickness of thin plate reaches a certain value, and the shear effect needs to be considered. In addition, the increase in stiffness can also be achieved by bonding laminated plates and sandwich plates, which is not suitable for industrial production and cannot meet engineering requirements [1,2]. Moreover, subsequent studies have found that the production process of laminated plates and sandwich plates is relatively complex and prone to degumming or dislocation, which is not conducive to the use in industrial production [3,4,5].

As a new type of lightweight structure, the convex-concave plate, made by stamping and rolling can avoid the shortcomings of the degumming, dislocation, and complicated process, resulting in high industrial application value and reliability. Although its thickness is very small, it has light weight, high strength, and high stiffness. This is because the convexities of the plate changes the plate structure after stamping or rolling, and the stiffness is greatly enhanced. Furthermore, the convexities can absorb a lot of energy in the process of impact vibration, and the loss of the plate can be greatly reduced [6]. This kind of plate also has high heat dissipation efficiency and excellent sound insulation and heat insulation performance, which cannot be achieved by the sandwich plate and laminated plate. In fact, it has been used as the plate heat exchanger as shown in Figure 1.

Due to the existence of the convexities, the energy absorption capacity is obviously improved and the service life is greatly enhanced [7,8,9]. Haldar et al. [10] focused on studying the influence of the thickness of convexities on the energy absorption and obtained the optimal energy absorption properties of the convex-concave plate by comparing the experimental and finite element simulation results. Sashikumar [11] tested the energy absorption capacity of an aluminum egg-box plate in commercial vehicles and proposed that the natural constraints within the geometric structure of egg-box plates have a positive impact on their energy absorption capacity. Akisanya and Fleck [12] discussed the dependence of strength and energy absorption of egg-box material on the geometric shape by using the finite element prediction method and determined the variation of the stiffness and strength of egg-box material with relative density. Ma et al. [13] concluded that the periodic origami structure has better energy absorption capacity by comparing the dynamic analysis and experimental and numerical results.

In addition to the traditional single-layer convex-concave plate, the laminated convex-concave plate has also been widely investigated. Chang et al. [14,15,16] adopted the experimental method to study the energy absorbing ability and deformation process of two kinds of sinusoidal convex-concave plate reinforced by fiber materials during quasi-static compression. Zhang et al. [17] tested the performance of egg-box panel stuffed whipple shield from impact area, cell size and the axial offset etc. At present, experiments and finite element simulation are mostly used to investigate the energy absorption capacity, failure, and heat transfer performance of concave-convex plates.

The laminated composite plate by nature has two dimensions larger than the thickness dimension by an order of magnitude. In recent years, the variational asymptotic method (VAM) has been used to strictly split the plate/shell problem into the through-the-thickness linear analysis (1-D analysis) and 2D nonlinear plate analysis by using the small parameter of width-to-thickness ratio. This method combines advantages of both asymptotic and variational methods, and considers all possible deformations during problem formulations while avoiding any kinematic assumptions. Atilgan and Hodges [18,19] developed VAM for laminated plates in which each lamina exhibits monoclinic symmetry about its own midplane. Sutyrin and Hodges [20] extended VAM to laminated plate, the material properties of which varied through the thickness and for which each lamina was orthotropic. Later, Sutyrin [21] developed linear, asymptotically correct theories for inhomogeneous orthotropic plates, for example, laminated plates with orthotropic laminae. Yu et al. [22] developed an accurate stress/strain recovery procedure for laminated plates that can be implemented in standard finite element programs. Kamineni and Burela [23] developed constraint method for laminated composite flat stiffened panel using VAM. Further, Zhong and Yu [24,25,26,27,28] successfully constructed the equivalent model of various composite structures by using this method.

The composite conical convex-concave plate (CCCP) studied in this article is a new type of convex-concave plate. Most of the research at this stage is limited to the manufacturing process, and its equivalent mechanical properties have not been investigated, especially the natural vibration and stability characteristics. Therefore, it is necessary to study the equivalent stiffness of the orthotropic convex-concave plate, which lays a foundation for studying the deformation of convex-concave plates under applied load and provides a theoretical basis for the reasonable design of its geometric parameters.

In this work, a VAM-based equivalent plate model is developed to predict the effective performance of CCCP. The organization of this paper is as follows. The theoretical formulation for the VAM-based 2D equivalent plate model (2D-EPM) is deduced in Section 2. Numerical examples of static displacement, global buckling, and free-vibration analysis of CCCP are used in Section 3 to verify the accuracy and effectiveness of 2D-EPM. Section 4 investigates the influences of layup configurations and geometric parameters on the effective performances of CCCP by using the 2D-EPM. In Section 5, the effective performance of bidirectional CCCP and CCCP is compared. Finally, some conclusions are drawn in Section 6.

## 2. Theoretical Foundation

### 2.1. Equivalent Plate Modeling of CCCP

The equivalent model of CCCP can be established from the perspective of energy concept. Due to the periodicity of the CCCP along the two axes of the plane, a square plate with conical convexity is taken as a typical unit cell as shown in Figure 2b. The length, height, and thickness of the unit cell are *l*, *h*, and *t*, respectively. That is, the analysis of the original CCCP is decomposed into a microscopic analysis of the unit cell (providing effective plate properties) and macro analysis of 2D-EPM as shown in Figure 2c.

To facilitate the derivation, two groups of coordinates are introduced: the macro coordinates, $x_{i}$, describing the original structure, and the micro coordinates, $y_{i}$, describing the unit cell. The macro coordinate $x_{3}$ is perpendicular to the in-plane coordinates $x_{1}$ and $x_{2}$, and the origin of the macro coordinate is located at the center of the plate as shown in Figure 2a. Because the micro size of the unit cell is much smaller than the macro size of the plate, the micro coordinates $y_{i}=x_{i}/\xi$ (1/ξ is a small parameter) are used to describe unit cell. For the equivalent plate model, the function of the original CCCP can be expressed as a function defined along the reference plane $x_{1}-x_{2}$ ($x_{3}$ disappears), and its partial derivative is
(1)$$\frac{\partial f\left(x_{\alpha };y_{i}\right)}{\partial x_{\alpha }}={\left.\frac{\partial f\left(x_{\alpha };y_{i}\right)}{\partial x_{\alpha }}\right|}_{y_{i}=const}+{\left.\frac{1}{\xi }\frac{\partial f\left(x_{\alpha };y_{i}\right)}{\partial y_{i}}\right|}_{x_{\alpha }=const}\equiv f_{,\alpha }+\frac{1}{\xi }f_{;i}$$
where *i* takes values of 1, 2, and 3; α takes values 1 and 2.

To establish the VAM-based equivalent model of CCCP, the 2D plate variables should be used to represent the 3D displacements of the original CCCP as
(2)$$\begin{array}{c} u_{1}\left(x_{\alpha };y_{i}\right)=\underline{v_{1}\left(x_{1},x_{2}\right)-\xi y_{3}v_{3,1}\left(x_{1},x_{2}\right)}+\xi w_{1}\left(x_{\alpha };y_{i}\right)\\ u_{2}\left(x_{\alpha };y_{i}\right)=\underline{v_{2}\left(x_{1},x_{2}\right)-\xi y_{3}v_{3,2}\left(x_{1},x_{2}\right)}+\xi w_{2}\left(x_{\alpha };y_{i}\right)\\ u_{3}\left(x_{\alpha };y_{i}\right)=\underline{v_{3}\left(x_{1},x_{2}\right)}+\xi w_{3}\left(x_{\alpha };y_{i}\right) \end{array}$$
where $u_{i}$ and $v_{i}$ represent the displacement of the original three-dimensional plate and two-dimensional equivalent plate, respectively, and $w_{i}$ denotes the fluctuating function used to describe the deformation that cannot be described by the classical plate theory.

The underlined terms in Equation (Equation 2) are the deformations generated by the reference plane, which should meet the following requirements:(3)$$hv_{\alpha }\left(x_{\alpha }\right)=\langle u_{\alpha }\rangle +\langle \xi y_{3}\rangle v_{3,\alpha },hv_{3}\left(x_{\alpha }\right)=\langle u_{3}\rangle$$
where the angle brackets denote the integral over the unit cell.

The definitions in Equation (Equation 3) introduce the constraints on the fluctuating functions as
(4)$$\langle \xi w_{i}\rangle =0$$

The strain field can be obtained from linear elasticity theory as
(5)$$\Gamma_{ij}=\frac{1}{2}\left(\frac{\partial u_{i}}{\partial x_{j}}+\frac{\partial u_{j}}{\partial x_{i}}\right)$$

The corresponding 3D linear strain field can be obtained by substituting Equation (Equation 2) into Equation (Equation 5), and neglecting higher-order terms,
(6)$$\begin{array}{c} \Gamma_{11}=\varepsilon_{11}+\xi y_{3}\kappa_{11}+w_{1,1}\\ 2\Gamma_{12}=2\varepsilon_{22}+2\xi y_{3}\kappa_{12}+w_{1,2}+w_{2,1}\\ \Gamma_{22}=\varepsilon_{22}+\xi y_{3}\kappa_{22}+w_{2,2}\\ 2\Gamma_{13}=w_{1,3}+w_{3,1}\\ 2\Gamma_{23}=w_{2,3}+w_{3,2}\\ \Gamma_{33}=w_{3,3} \end{array}$$
where the 2D plate strains $\varepsilon_{\alpha \beta }$ and $\kappa_{\alpha \beta }$ of the 2D-EPM are defined as
(7)$$\varepsilon_{\alpha \beta }\left(x_{1},x_{2}\right)=\frac{1}{2}\left(v_{\alpha ,\beta }+v_{\beta ,\alpha }\right),\;\kappa_{\alpha \beta }\left(x_{1},x_{2}\right)=-v_{3,\alpha \beta }$$

With the aid of matrix representation, the 3D strain field can be expressed as
(8)$$\begin{array}{c} \Gamma_{e}={\left[\Gamma_{11}\;\Gamma_{22}\;2\Gamma_{12}\right]}^{T}=\varepsilon +x_{3}\kappa +\partial_{e}w_{\left|\right|}\\ 2\Gamma_{s}={\left[2\Gamma_{13}\;2\Gamma_{23}\right]}^{T}=w_{\left|\right|}+\partial_{t}w_{3}\\ \Gamma_{t}=\Gamma_{33}=w_{3,3} \end{array}$$
where $\Gamma_{e},\Gamma_{s},\Gamma_{t}$ are strain components of 3D-FEM, respectively; ${\left(\right)}_{\left|\right|}={\left[{\left(\right)}_{1}\;{\left(\right)}_{2}\right]}^{T}$, $\varepsilon ={\left[\varepsilon_{11}\;2\varepsilon_{12}\;\varepsilon_{22}\right]}^{T}$, $\kappa ={\left[\kappa_{11}\;\kappa_{12}+\kappa_{21}\;\kappa_{22}\right]}^{T}$, and
(9)$$\partial_{e}=\left[\begin{array}{cc} {\left(\right)}_{,1} & 0\\ {\left(\right)}_{,2} & {\left(\right)}_{,1}\\ 0 & {\left(\right)}_{,2} \end{array}\right],\partial_{t}=\left\{\begin{array}{c} {\left(\right)}_{,1}\\ {\left(\right)}_{,2} \end{array}\right\}$$

The geometric structure of the element can be divided into five parts shown in Figure 3 for easy integration, and the strain energy of unit cell can be written as
(10)$$\begin{array}{c} U=\int_{-l_{3}/2}^{l_{3}/2}\int_{-\left(l_{3}/2+l_{1}+l_{2}\right)}^{l_{3}/2+l_{1}+l_{2}}\int_{0}^{h}\Gamma_{C}^{T}D_{C}\Gamma_{C}dy_{1}dy_{2}dy_{3}\\ \;+2\int_{l_{3}/2}^{l_{3}/2+l_{2}}\int_{-\left(l_{3}/2+l_{1}+l_{2}\right)}^{l_{3}/2+l_{1}+l_{2}}\int_{0}^{h}\Gamma_{B}^{T}D_{B}\Gamma_{B}dy_{1}dy_{2}dy_{3}\\ \;+2\int_{l_{3}/2+l_{2}}^{l_{3}/2+l_{2}+l_{1}}\int_{-\left(l_{3}/2+l_{1}+l_{2}\right)}^{l_{3}/2+l_{1}+l_{2}}\int_{-t}^{0}\Gamma_{A}^{T}D_{A}\Gamma_{A}dy_{1}dy_{2}dy_{3} \end{array}$$

Equation (Equation 10) can be briefly expressed as
(11)$$U=\frac{1}{2}\langle \Gamma^{T}D\Gamma \rangle =\frac{1}{2}\langle {\left\{\begin{array}{c} \Gamma_{e}\\ 2\Gamma_{s}\\ \Gamma_{t} \end{array}\right\}}^{T}\left[\begin{array}{ccc} D_{e} & D_{es} & D_{et}\\ D_{es}^{T} & D_{s} & D_{st}\\ D_{et}^{T} & D_{st}^{T} & D_{t} \end{array}\right]\left\{\begin{array}{c} \Gamma_{e}\\ 2\Gamma_{s}\\ \Gamma_{t} \end{array}\right\}\rangle$$
where $D_{e},D_{es},D_{et},D_{s},D_{st}$, and $D_{t}$ denote the sub-matrices of the three-dimensional 6×6 material matrix.

The virtual work done due to applied loads can be expressed as
(12)$$\delta {\bar{W}}_{3D}=\delta {\bar{W}}_{2D}+\delta {\bar{W}}^{*}$$
where $\delta {\bar{W}}_{2D}$ and $\delta {\bar{W}}^{*}$ can be defined as
(13)$$\delta {\bar{W}}_{2D}=\langle p_{i}\delta v_{i}+q_{\alpha }\delta v_{3,\alpha }\rangle ,\delta {\bar{W}}^{*}=\langle \langle f_{i}\delta w_{i}\rangle +\tau_{i}\delta w_{i}^{+}+\beta_{i}\delta w_{i}^{-}\rangle$$
where ${\left(\cdot \right)}^{+}={\left.(\cdot )\right|}_{x_{3}=t/2}$ and ${\left(\cdot \right)}^{-}={\left.(\cdot )\right|}_{x_{3}=-t/2}$ denote the items acting on the top and bottom surfaces, respectively; $f_{i}$ is the body force; $\tau_{i}$ and $\beta_{i}$ denote the traction forces on the top and bottom surface, respectively; and $p_{i}=\langle f_{i}\rangle +\tau_{i}+\beta_{i},q_{\alpha }=h/2\left(\beta_{\alpha }-\tau_{\alpha }\right)-\langle x_{3}f_{\alpha }\rangle$.

The total potential energy of unit cell can be expressed as
(14)$$\delta \Pi =\delta U-\delta W^{*}=\frac{1}{2}\delta \langle \Gamma^{T}D\Gamma \rangle -\langle \langle f_{i}\delta w_{i}\rangle +\tau_{i}\delta w_{i}^{+}+\beta_{i}\delta w_{i}^{-}\rangle$$

### 2.2. Dimension Reduction of CCCP

To solve the unknown fluctuating function $w_{i}$ by using VAM, the order of each term in Equation (Equation 14) must be evaluated as
(15)$$\begin{array}{c} \Gamma_{ij}∼\varepsilon_{\alpha \beta }∼h\kappa_{\alpha \beta }∼n,w_{i}∼hn,w_{||,\alpha }∼w_{3,\alpha }∼\frac{h}{L}n\\ w_{||,3}∼w_{3,3}∼n,hf_{\alpha }∼\alpha_{\alpha }∼\beta_{\alpha }∼\mu \frac{h}{L}n,hf_{3}∼\alpha_{3}∼\beta_{3}∼\mu {\left(\frac{h}{L}\right)}^{2} \end{array}$$
where *n* and μ are the order of the minimum strain and material properties, respectively, and *L* is the length of the plate.

#### 2.2.1. Zeroth-Order Approximation

The explicit expression of Equation (Equation 14) can be expressed as
(16)$$\begin{array}{c} 2\Pi =\langle \underline{\underline{{\left(\varepsilon +x_{3}\kappa \right)}^{T}D_{e}\left(\varepsilon +x_{3}\kappa \right)}}+\underline{2{\left(\varepsilon +x_{3}\kappa \right)}^{T}D_{e}\partial_{e}w_{||,\alpha }}\\ \;+\underline{2{\left(\partial_{e}w_{||,\alpha }\right)}^{T}D_{e}\partial_{e}w_{\left|\right|}}+2{\left(\varepsilon +x_{3}\kappa \right)}^{T}D_{es}w_{||,3}+\underline{2{\left(\varepsilon +x_{3}\kappa \right)}^{T}D_{es}\partial_{t}w_{3,\alpha }}\\ \;+\underline{2{\left(\partial_{e}w_{||,\alpha }\right)}^{T}D_{es}\left(w_{||,3}+\partial_{t}w_{3,\alpha }\right)}+2{\left(\varepsilon +x_{3}\kappa \right)}^{T}D_{et}w_{3,3}\\ \;+\underline{2{\left(\partial_{e}w_{||,\alpha }\right)}^{T}D_{et}w_{3,3}}+w_{||,3}^{T}D_{s}w_{||,3}+\underline{2w_{||,3}^{T}D_{s}\partial_{t}w_{3,\alpha }}+\underline{2{\left(\partial_{t}w_{3,\alpha }\right)}^{T}D_{s}\partial_{t}w_{3,\alpha }}\\ \;+2w_{||,3}^{T}D_{st}w_{3,3}+\underline{2{\left(\partial_{t}w_{3,\alpha }\right)}^{T}D_{st}w_{3,3}}+D_{t}w_{3,3}^{2}\rangle \\ \;-\underline{2\left(\langle f_{i}^{T}w_{i}\rangle +\tau_{i}^{T}w_{i}^{T}+\beta_{i}^{T}w_{i}^{T}\right)} \end{array}$$

The fluctuating function can be constrained by introducing the Lagrangian multiplier $\lambda_{i}$, such as
(17)$$\delta \left(\Pi +\lambda_{i}\langle w_{i}\rangle \right)=0$$

The fluctuating function can be obtained by solving the following zeroth-order approximate variational statement after removing the underline and double underline items from Equation (Equation 16),
(18)$$\langle \begin{array}{c} \left[{\left(\varepsilon +x_{3}\kappa \right)}^{T}D_{es}+w_{||,3}^{T}D_{s}+w_{3,3}^{T}D_{st}^{T}\right]\delta w_{||,3}\\ +\lambda_{i}\delta w_{i}+\left[{\left(\varepsilon +x_{3}\kappa \right)}^{T}D_{et}+w_{||,3}^{T}D_{st}+w_{3,3}^{T}D_{t}\right]\delta w_{3,3} \end{array}\rangle =0$$

This results in the following Euler–Lagrange equations:(19)$$\begin{array}{c} {\left[{\left(\varepsilon +x_{3}\kappa \right)}^{T}D_{es}+w_{||,3}^{T}D_{s}+w_{3,3}^{T}D_{st}^{T}\right]}_{,3}=\lambda_{\left|\right|}\\ {\left[{\left(\varepsilon +x_{3}\kappa \right)}^{T}D_{et}+w_{||,3}^{T}D_{st}+w_{3,3}^{T}D_{t}\right]}_{,3}=\lambda_{3} \end{array}$$
where $\lambda_{\left|\right|}={\left[\lambda_{1}\;\lambda_{2}\right]}^{T}$ and $\lambda_{3}$ are Lagrange multipliers corresponding to the constraint components of $w_{\left|\right|}$ and $w_{3}$.

The boundary conditions are
(20)$$\begin{array}{c} {\left[{\left(\varepsilon +x_{3}\kappa \right)}^{T}D_{es}+w_{||,3}^{T}D_{s}+w_{3,3}D_{st}^{T}\right]}^{+/-}=0\\ {\left[{\left(\varepsilon +x_{3}\kappa \right)}^{T}D_{et}+w_{||,3}^{T}D_{st}+w_{3,3}D_{t}\right]}^{+/-}=0 \end{array}$$
where the superscript “+/−” denotes the items applied on the top and bottom surface of the plate.

We can solve $w_{i}$ by substituting Equation (Equation 20) back into Equation (Equation 19):(21)$$w_{\left|\right|}={\langle -\left(\varepsilon +x_{3}\kappa \right){\bar{D}}_{es}D_{s}^{-1}\rangle }^{T},w_{3}=\langle -\left(\varepsilon +x_{3}\kappa \right){\bar{D}}_{et}D_{t}^{-1}\rangle$$
where
(22)$${\bar{D}}_{es}=D_{es}-{\bar{D}}_{et}D_{st}^{T}{\bar{D}}_{t}^{-1},{\bar{D}}_{et}=D_{et}-D_{es}D_{s}^{-1}D_{st},{\bar{D}}_{t}=D_{t}-D_{sl}^{T}D_{s}^{-1}D_{st}$$

Substituting Equation (Equation 22) into Equation (Equation 18), we obtain the first approximation of the strain energy as
(23)$$U_{2D}=\frac{1}{2}\langle {\left(\varepsilon +x_{3}\kappa \right)}^{T}{\bar{D}}_{e}\left(\varepsilon +x_{3}\kappa \right)\rangle =\frac{1}{2}{\left\{\begin{array}{c} \varepsilon \\ \kappa \end{array}\right\}}^{T}\left[\begin{array}{cc} A & B\\ B^{T} & D \end{array}\right]\left\{\begin{array}{c} \varepsilon \\ \kappa \end{array}\right\}$$
where
(24)$$\begin{array}{c} A=\langle {\bar{D}}_{e}\rangle ,B=\langle x_{3}{\bar{D}}_{e}\rangle ,D=\langle x_{3}^{2}{\bar{D}}_{e}\rangle ,\\ {\bar{D}}_{e}=D_{e}-{\bar{D}}_{es}D_{s}^{-1}D_{es}^{T}-{\bar{D}}_{et}D_{et}^{T}/{\bar{D}}_{t} \end{array}$$

For the zeroth-order approximate, the 3D strain field can be recovered by using Equation (Equation 6) as
(25)$$\Gamma_{e}^{0}=\varepsilon +x_{3}\kappa ,\;2\Gamma_{s}^{0}=-w_{||,3},\;\Gamma_{t}^{0}=w_{3,3}$$

The local stress field can be recovered as
(26)$$\begin{array}{c} \sigma_{e}^{0}={\left[\sigma_{11}^{0}\;\sigma_{12}^{0}\;\sigma_{22}^{0}\right]}^{T}=\bar{D_{e}}\left(\varepsilon +x_{3}\kappa \right)\\ \sigma_{s}^{0}={\left[\sigma_{13}^{0}\;\sigma_{23}^{0}\right]}^{T}=0\\ \sigma_{t}^{0}=\sigma_{33}^{0}=0 \end{array}$$

The resultant stress of the plate can be defined as
(27)$$\begin{array}{c} \frac{\partial U_{2D}}{\partial \varepsilon_{11}}=N_{11},\;\frac{\partial U_{2D}}{\partial 2\varepsilon_{12}}=N_{12},\;\frac{\partial U_{2D}}{\partial \varepsilon_{22}}=N_{22}\\ \frac{\partial U_{2D}}{\partial \kappa_{11}}=M_{11},\;\frac{\partial U_{2D}}{\partial 2\kappa_{12}}=M_{12},\;\frac{\partial U_{2D}}{\partial \kappa_{22}}=M_{22} \end{array}$$

The constitutive relation of CCCP can be obtained by connecting the internal stress, strain, and curvature of the plate as
(28)$$\left\{\begin{array}{c} N_{11}\\ N_{22}\\ N_{12}\\ M_{11}\\ M_{22}\\ M_{12} \end{array}\right\}=\left[\begin{array}{cccccc} A_{11} & A_{12} & A_{16} & B_{11} & B_{12} & B_{16}\\ A_{12} & A_{22} & A_{26} & B_{12} & B_{22} & B_{26}\\ A_{16} & A_{26} & A_{66} & B_{16} & B_{26} & B_{66}\\ B_{11} & B_{12} & B_{16} & D_{11} & D_{12} & D_{16}\\ B_{12} & B_{22} & B_{26} & D_{12} & D_{22} & D_{26}\\ B_{16} & B_{26} & B_{66} & D_{16} & D_{26} & D_{66} \end{array}\right]\left\{\begin{array}{c} \varepsilon_{11}\\ \varepsilon_{22}\\ 2\varepsilon_{12}\\ \kappa_{11}\\ \kappa_{22}\\ 2\kappa_{12} \end{array}\right\}$$

#### 2.2.2. First-Order Approximation

It can be seen from Equation (Equation 26) that only in-plane stresses $\sigma_{e}$ can be obtained. The next order approximation is needed to obtain the out-of-plane stresses. The zeroth-order warping functions can be simply perturbed as
(29)$$w_{\left|\right|}={\bar{v}}_{\left|\right|},\;w_{3}={\bar{v}}_{3}+D_{⊥}\chi$$
where $\chi ={\left[\varepsilon \;\kappa \right]}^{T},D_{⊥}=\left[-\frac{D_{\mathrm{et}}^{T}}{D_{t}}\;-x_{3}\frac{D_{\mathrm{et}}^{T}}{D_{t}}\right]$, ${\bar{v}}_{\left|\right|}$ and ${\bar{v}}_{3}$ denote in-plane and out-of-plane perturbed fluctuating functions, respectively.

The leading terms for the first-order approximation of variational statement can be obtained by substituting Equation (Equation 29) back into Equation (Equation 14) as
(30)$$\begin{array}{c} 2\Pi_{1}=\langle {\bar{v}}_{||,3}^{T}D_{s}{\bar{v}}_{||,3}+D_{t}{\bar{v}}_{3,3}^{2}+2{\bar{v}}_{\left|\right|}^{T}C_{||,3}\chi_{,\alpha }+2{\bar{v}}_{\left|\right|}^{T}D_{s}\partial_{t}D_{⊥}\chi_{,\alpha }-2{\bar{v}}_{\left|\right|}^{-T}p_{\left|\right|}\rangle \\ -2v_{\left|\right|}^{-T}\tau_{\left|\right|}-2v_{\left|\right|}^{T}\beta_{\left|\right|} \end{array}$$

The Euler–Lagrange equation of Equation (Equation 30) is
(31)$${\left(D_{s}{\bar{v}}_{||,3}+D_{s}\partial_{t}D_{⊥}\chi_{,\alpha }\right)}_{,3}=C_{||,3}\partial_{t}D_{⊥}\chi_{,\alpha }+g_{,3}+\lambda_{\left|\right|}$$
where $C_{\left|\right|}=-\partial_{e}^{T}\left[D_{\left|\right|}\;x_{3}D_{\left|\right|}\right],g_{,3}=-p_{\left|\right|}$.

It can be easily observed that ${\bar{v}}_{3}$ is decoupled from ${\bar{v}}_{\left|\right|}$ and only has a trivial solution. The solution of ${\bar{v}}_{\left|\right|}$ can be obtained from Equation (Equation 31) as
(32)$${\bar{v}}_{\left|\right|}=\left({\bar{C}}_{\left|\right|}+L_{\alpha }\right)\chi_{,\alpha }+\bar{g}$$
where
(33)$$\begin{array}{c} {\bar{C}}_{||,3}=D_{s}^{-1}C_{\left|\right|},\;\langle {\bar{C}}_{\left|\right|}\rangle =0,\;{\bar{g}}_{3}=D_{s}^{-1}\bar{g},\;\langle \bar{g}\rangle =0\\ L_{\alpha }\chi_{,\alpha }=c_{\left|\right|}/h,\;{\bar{C}}_{\left|\right|}=C_{\left|\right|}+\frac{x_{3}}{h}D_{\alpha }^{\mp }-\frac{1}{2}D_{\left|\right|}^{\pm }-D_{s}e_{\alpha }D_{⊥}\\ \bar{g}=g+\frac{x_{3}}{h}g^{\mp }-\frac{1}{2}g^{\pm } \end{array}$$
with ${\left(\right)}^{\pm }={\left(\right)}^{+}+{\left(\right)}^{-},{\left(\right)}^{\mp }={\left(\right)}^{-}-{\left(\right)}^{+}$.

So far, the asymptotic correction solution of the strain energy per unit area of the plate in the first-order approximation is
(34)$$2\Pi_{1}=\chi^{T}{\bar{D}}_{e}\chi +\chi_{,\alpha }^{T}B_{\alpha \beta }\chi_{,\beta }-2\chi^{T}F$$
where
(35)$$\begin{array}{c} B_{\alpha \beta }=\langle D_{s_{\alpha \beta }}D_{⊥}^{T}D_{⊥}-{\bar{C}}_{\alpha }^{T}D_{s}^{-1}{\bar{C}}_{\beta }\rangle +L_{\alpha }{\left(\langle \bar{p}\rangle +\langle p_{\left|\right|}\rangle \right)}_{,\alpha }\\ F=\langle D_{⊥}^{T}p_{3}\rangle -\langle {\bar{C}}_{∥}^{T}D_{s}^{-1}g_{s}\rangle -L_{\alpha }{\left(\langle \bar{p}\rangle +\langle p_{∥}\rangle \right)}_{,\alpha } \end{array}$$

The variational statement of Equation (Equation 34) governs the macroscopic behavior of the plate, and it only involves the 2-D field variables in terms of the macro-coordinates $x_{1}$ and $x_{2}$. Therefore, 2D-EPM can replace the original CCCP for the global analysis.

#### 2.2.3. Recovery 3D Local Fields

The fidelity of the equivalent model should be evaluated based on how well it can predict the 3D local fields for the original 3D structure. Therefore, it is necessary to provide a recovery relationship to complete the equivalent model so that the results is comparable to those of the original 3D model.

The 3D displacement field can be recovered by using Equation (Equation 2) as
(36)$$u_{i}=v_{i}+x_{3}\left(C_{3i}-\delta_{3i}\right)+C_{ji}w_{j}$$
where $u_{i}$ and $v_{i}$ are displacements of three-dimensional plate and two-dimensional equivalent plate, respectively, and $C_{ji}$ is the cosine component of the transformation matrix from macro coordinates to micro coordinates.

The 3D strain field can be recovered from Equation (Equation 8) as
(37)$$\Gamma_{e}=\varepsilon +x_{3}\kappa ,\;2\Gamma_{s}={\bar{v}}_{||3}+\partial_{t}D_{⊥,\alpha },\;\Gamma_{t}=D_{⊥,3}\chi$$

At last, the 3D stress field can be recovered as
(38)$$\begin{array}{c} \sigma_{e}={\left[\sigma_{11}\;\sigma_{12}\;\sigma_{22}\right]}^{T}=D_{\left|\right|}\left(\varepsilon +x_{3}\kappa \right)+D_{e}\partial_{e}{\bar{v}}_{||,\alpha }\\ 2\sigma_{s}={\left[\sigma_{13}\;\sigma_{23}\right]}^{T}=D_{s}\left({\bar{v}}_{||,3}+\partial_{t}\chi_{,\alpha }\right)\\ \sigma_{t}=\sigma_{33}=D_{et}^{T}\partial_{e}{\bar{v}}_{||,\alpha } \end{array}$$

### 2.3. Free-Vibration Analysis of CCCP Using 2D-EPM

The elastic curved surface differential equation of 2D-EPM under lateral load τ is
(39)$$D_{11}\frac{\partial^{4}v}{\partial x_{1}^{4}}+2D_{66}\frac{\partial^{4}v}{\partial x_{1}^{2}\partial x_{2}^{2}}+D_{22}\frac{\partial^{4}v}{\partial x_{2}^{4}}=\tau$$

It is assumed that the plate is in equilibrium under the lateral load τ at a certain moment, and the deflection of 2D-EPM at the equilibrium position is
(40)$$v_{a}=v_{a}\left(x_{1},x_{2}\right)$$

It is well known that 2D-EPM will be in a static state after moving for a period of time. Then, 2D-EPM will vibrate freely after the appropriate disturbing force is removed. The deflection of 2D-EPM at a certain moment under the condition of free vibration is
(41)$$v_{t}=v_{t}\left(x_{1},x_{2},t\right)$$

In addition to the lateral load τ, there is also the inertia force $\tau_{j}$ after the disturbing force is removed, and the equilibrium equation becomes
(42)$$D_{11}\frac{\partial^{4}v_{t}}{\partial x_{1}^{4}}+2D_{66}\frac{\partial^{4}v_{t}}{\partial x_{1}^{2}\partial x_{2}^{2}}+D_{22}\frac{\partial^{4}v_{t}}{\partial x_{2}^{4}}=\tau +\tau_{j}$$

The acceleration of 2D-EPM is $\frac{\partial^{2}v}{\partial t^{2}}$. If the density and thickness of 2D-EPM are, respectively, $\rho_{0}$ and $\delta_{0}$, then the inertia force $\tau_{j}$ is
(43)$$\tau_{j}=-\rho_{0}\delta_{0}\frac{\partial^{2}v}{\partial t^{2}}$$

Substituting Equation (Equation 43) into Equation (Equation 42), and Equation (Equation 40) into Equation (Equation 39), the final differential equation can be obtained after subtracting Equation (Equation 39) from Equation (Equation 42) as
(44)$$D_{11}\frac{\partial^{4}\left(v_{t}-v_{a}\right)}{\partial x_{1}^{4}}+2D_{66}\frac{\partial^{4}\left(v_{t}-v_{a}\right)}{\partial x_{1}^{2}\partial x_{2}^{2}}+D_{22}\frac{\partial^{4}\left(v_{t}-v_{a}\right)}{\partial x_{2}^{4}}=-\rho_{0}\delta_{0}\frac{\partial^{2}\left(v_{t}-v_{a}\right)}{\partial t^{2}}$$

The deflection of 2D-EPM satisfies the following conditions at any time:(45)$$v=v_{t}-v_{a}$$

Equation (Equation 44) can be simplified as
(46)$$D_{11}\frac{\partial^{4}v}{\partial x_{1}^{4}}+2D_{66}\frac{\partial^{4}v}{\partial x_{1}^{2}\partial x_{2}^{2}}+D_{22}\frac{\partial^{4}v}{\partial x_{2}^{4}}+\rho_{0}\delta_{0}\frac{\partial^{2}v}{\partial t^{2}}=0$$
and its general solution is
(47)$$v=\sum_{m=1}^{\infty }v_{m}=\sum_{m=1}^{\infty }\left(A_{m}\cos\omega t+B_{m}\sin\omega t\right)W\left(x,y\right)$$
where *W* is the free-vibration shape function and ω is the natural frequency (Hz).

As each natural frequency of 2D-EPM corresponds to its shape function, the differential equation can be obtained by substituting any of the natural frequency into Equation (Equation 46):(48)$$D_{11}\frac{\partial^{4}W}{\partial x_{1}^{4}}+2D_{66}\frac{\partial^{4}W}{\partial x_{1}^{2}\partial x_{2}^{2}}+D_{22}\frac{\partial^{4}W}{\partial x_{2}^{4}}=-\omega^{2}\rho_{0}\delta_{0}W$$

The nonzero solution satisfying *W* can be obtained as
(49)$$\omega^{2}=\frac{D_{11}\frac{\partial^{4}W}{\partial x_{1}^{4}}+D_{22}\frac{\partial^{4}W}{\partial x_{2}^{4}}+2D_{66}\frac{\partial^{4}W}{\partial x_{1}^{2}\partial x_{2}^{2}}}{\rho_{0}\delta_{0}W}$$

The free-vibration shape function of the four-side simple supporting plate can be defined as
(50)$$W=\sin\frac{m\pi x_{1}}{a}\sin\frac{n\pi x_{2}}{b}$$
where *m* and *n* are half-wave numbers along $x_{1}$ and $x_{2}$ directions, respectively; *a* and *b* are the side lengths of 2D-EPM, respectively.

Substituting Equation (Equation 50) back into Equation (Equation 49), we can obtain
(51)$$\omega^{2}=\frac{D_{11}{\left(\frac{m\pi }{a}\right)}^{4}+D_{22}{\left(\frac{n\pi }{b}\right)}^{4}+2D_{66}{\left(\frac{m\pi }{a}\right)}^{2}{\left(\frac{n\pi }{b}\right)}^{2}}{\rho_{0}\delta_{0}}$$

Therefore, the natural frequencies of 2D-EPM can be obtained as
(52)$$\omega =\pi^{2}\sqrt{\frac{D_{11}{\left(\frac{m}{a}\right)}^{4}+D_{22}{\left(\frac{n}{b}\right)}^{4}+2D_{66}{\left(\frac{m}{a}\right)}^{2}{\left(\frac{n}{b}\right)}^{2}}{\rho_{0}\delta_{0}}}$$

## 3. Model Validation

In this section, the accuracy and effectiveness of the equivalent model are verified by comparing with the results of static displacement, free-vibration, and global buckling analysis of the three-dimensional finite element model (3D-FEM).

The dimensions of unit cell are l=30 mm, h=10 mm, t=1 mm, $l_{1}=5$ mm, $l_{2}=5$ mm, and $l_{3}=10$ mm. The whole model of CCCP is the repetition of the unit cell in Figure 2b 20 times along $x_{1}$ and $x_{2}$ directions. The layup configuration of laminate is [0/45/90/−45/0]s, and each ply thickness equals 0.1 mm. The material used is a unidirectional carbon fiber/epoxy resin composite (T300/7901), and the lamina properties are $E_{11}=E_{22}$ = 48.36 GPa, $E_{33}$ = 12.74 GPa, $G_{12}$ = 11.37 GPa, $G_{13}$ = $G_{23}$ = 4.59 GPa, $v_{12}$ = 0.078, $v_{13}$ = $v_{23}$ = 0.397, ρ = 1670 $\mathrm{kg}/m^{3}$. The effective plate properties of CCCP obtained by present model is shown in Figure 4.

### 3.1. Static Deformation Analysis

To investigate the static deformation of CCCP under different boundary conditions, the four cases shown in Figure 5 are considered. Table 1 lists the static displacements of 2D-EPM and 3D-FEM when a concentrated load of 100 N is applied to the center of the plate. It can be seen that the displacement distribution of 2D-EPM agrees with that of 3D-FEM under different boundary conditions. The boundary condition has a great influence on the relative errors. The smaller the boundary constraint is, the larger the displacement error is. However, the maximum error is less than 6% in Case 4, indicating the equivalent stiffness obtained by VAM is accurate, and the present 2D-EPM can well reflect the static displacement of CCCP under different boundary conditions.

### 3.2. Recovery 3D Field

As shown in Equations (Equation 36)–(Equation 38), the global displacements and strains of 2D-EPM should be imputed into the corresponding unit cell to recover the local field distribution. Note that most of the simplified models cannot predict the accurate local field distribution.

It can be seen from Figure 6 that the distribution of local stress within the unit cell is not uniform. The values of $\sigma_{11}$ and $\sigma_{22}$ are larger on the conical convexity, while other stresses are mainly concentrated at the intersection between the flat plate and the conical convexity. That is, these stresses on the conical convexity are very small, and the conical convexity cannot completely resist the load. It can be reasonably explained that the plate fails at the intersection of the flat plate and the conical convexity.

Figure 7 shows the distribution of the local fields within the unit cell closest to the midpoint of the plate in Case 1 (CCCC). It is seen that the displacement within the unit cell is symmetrically distributed, and the minimum displacement is located at the edge of the flat plate, while the maximum displacement is located at the intersection of the flat plate and the conical convexity. The changes of $U_{1}$ and $U_{2}$ are opposite on both sides of the conical convexity, and the maximum and minimum values of $U_{1}$ and $U_{2}$ are also located close to the intersection of the flat plate and the conical convexity.

Figure 8 shows the local stress and displacement distributions along Path 1 predicted by 2D-EPM and 3D-FEM. It can be seen that the local stress and displacement distributions predicted by the two models are in good agreement, and the maximum error is less than 2%. The location of stress concentration and the change trend can be obtained directly according to the local field distribution, which provides a reference for the damage analysis of the structure.

### 3.3. Global Buckling Analysis

In this section, the linear buckling behavior of CCCP is predicted by 2D-EPM and 3D-FEM under the assumption of small displacements. Four sides of CCCP are simple-supported and two opposite sides are pressed. Table 2 shows that the global buckling modes of 2D-EPM are consistent with those of 3D-FEM. For example, there are one, two, three, and four half-wave along the horizontal direction in the first, second, third, and fourth mode shape of 2D-EPM and 3D-FEM, respectively. The maximum error of critical buckling load is 2.08% in the first order buckling mode, indicating the equivalent model based on VAM has high accuracy in buckling analysis of CCCP.

### 3.4. Free-Vibration Analysis

In this section, the natural frequencies and vibration mode shapes of CCCP are predicted by 2D-EPM and 3D-FEM. The geometry and boundary conditions are the same as those in Section 3.3. Table 3 shows that the vibration modes predicted by 2D-EPM are consistent with those predicted by 3D-FEM, and the maximum error of natural frequency is within 5%. The accuracy of 2D-EPM based on VAM is verified from the aspect of vibration characteristics.

### 3.5. Comparison of Calculation Efficiency

To further demonstrate the advantages of the model, the computational efficiencies of the three-dimensional FE model and two-dimensional equivalent model are compared as shown in Table 4. It can be observed that the present model is more time-efficient and cost-efficient than 3D-FEM in performing static, global buckling, and free-vibration analysis of CCCP. Note that the geometry of CCCP studied in this article is relatively simple, and the advantages of 2D-EPM can be reflected in more complex composite structures.

## 4. Parameter Analysis

### 4.1. Influence of Structural Parameters

The structural parameters mainly include $l_{1},l_{3},h$ and *t* as shown in Figure 3. In this section, the influence of structural parameters on the effective performance of CCCP is analyzed by using control variable method.

Figure 9a shows the influence of $l_{1}$ on the equivalent stiffness of CCCP. The parameters $l_{3}$ and *h* remain unchanged when considering the influence of $l_{1}$, so the total length of the unit cell is variable. It can be observed that the equivalent tensile stiffness of CCCP increases with increasing of $l_{1}$, while the equivalent bending stiffness gradually decreases nonlinearly. The main reason is that with the increase of the spacing between the conical convex part, the concave-convex characteristics of CCCP become less obvious, and the geometry of the structure is more similar to an ordinary flat plate.

Figure 9b shows that the first three buckling loads and natural frequencies of CCCP decrease with the increase of $l_{1}$. The critical buckling load of CCCP with $l_{1}$ = 2 mm is about three times that of $l_{1}$ = 10 mm, while the difference of natural frequency is about double, indicating that $l_{1}$ has a great influence on the buckling load and natural frequency of CCCP.

Figure 10a shows the influence of $l_{3}$ on the equivalent stiffness of CCCP. It can be observed that the width of convexity $l_{3}$ has little influence on the equivalent tensile and bending stiffness of CCCP. The curves of $A_{11}$ and $A_{22}$ decrease slightly, while the curve of $D_{11}$ shows an gentle and irregular change. The buckling loads and natural frequencies of CCCP shown in Figure 10b increase slightly with the increasing $l_{3}$. Therefore, the width of convexity $l_{3}$ has little influence on the effective performance of CCCP.

Figure 11a shows the influence of height *h* on the equivalent stiffness of CCCP. It can be observed that with the increase in convex height *h*, the equivalent tensile stiffness of CCCP decreases within a small range. The bending stiffness reaches the minimum value at *h* = 8 mm and then increases gradually with increasing of *h*. It can be seen form Figure 11b that the critical buckling load and natural frequency of CCCP increase to a certain extent with increasing of *h*, indicating the height of the convexity has a certain influence on the overall performance of the structure.

Figure 12a shows the influence of thickness *t* on the equivalent stiffness of CCCP. It can be seen that the equivalent tensile stiffness of CCCP increases linearly with the increasing thickness *t*, while the equivalent bending stiffness increases nonlinearly, which has a great influence on the buckling load and natural frequency of CCCP as shown in Figure 12b. The reason is that with the plate thickness *t* increases, the orthogonality of CCCP becomes non-obvious.

### 4.2. The Effect of Layup Configuration

#### 4.2.1. The Influence of Ply Angles

In this section, the influence of ply angles on the effective performance of CCCP is analyzed. The layup configuration of laminate is set as ${\left[0/\theta /0/\theta /0\right]}_{s}$, where the value of θ is from $0^{\circ }$ to $90^{\circ }$ with an interval of $15^{\circ }$.

Figure 13a shows the influence of ply angle on equivalent stiffness of CCCP. It can be observed that the curves of $A_{11}$ and $D_{11}$ show a nonlinear downward trend with the increase of the ply angles, while the curves of $A_{22}$ and $D_{22}$ show a small increase between ply angles of $0^{\circ }$ and $45^{\circ }$, and a nonlinear increase between ply angles of $45^{\circ }$ and $90^{\circ }$. It can be observed from Figure 13b that the influence of ply angle on buckling load is not obvious, while the second-order natural frequency reaches the maximum value when ply angles is $90^{\circ }$.

#### 4.2.2. The Influence of the Proportion of $0^{\circ }$ Ply

In this section, CCCP with the combination of $0^{\circ }$ and $45^{\circ }$ ply is adopted to study the influence of the proportion of $0^{\circ }$ ply on the effective stiffness, buckling, and natural frequencies. It can be observed from Figure 14a that the values of $A_{11}$ and $D_{11}$ gradually increase with the increase in the proportion of $0^{\circ }$ ply, while other stiffness gradually decrease. The buckling load of CCCP in Figure 14b gradually increases with the increase in the proportion of $0^{\circ }$ ply, and reaches its maximum value when the proportion of $0^{\circ }$ ply is 80%. The first two natural frequencies of CCCP increase and then decrease with increasing of the proportion of $0^{\circ }$ ply. Therefore, the buckling resistance of CCCP can be improved by optimizing the layup configuration in engineering application, so that the effective plate properties of CCCP can be brought into full play.

## 5. Bidirectional CCCP

The structure introduced in this section is a bidirectional CCCP, and the convex direction along each row and column of the structure is alternately arranged up and down periodically as shown in Figure 15, which can be used to improve heat transfer effect and reduce pressure drop. Each convexity and its periphery is regarded as a small element, and only four adjacent elements (2 × 2) need to be studied due to the periodicity of bidirectional CCCP.

The geometrical sizes of unit cell are: *h* = 10 mm, *t* = 1.0 mm, $l_{1}$ = 5 mm, $l_{2}$ = 5 mm, $l_{3}$ = 10 mm, periodic length *l* = 60 mm. The material properties and boundary conditions are the same as those in Section 3.1. The equivalent stiffness matrix of the bidirectional CCCP obtained by the present model is shown in Figure 16 for reference.

Table 5 shows the displacements of bidirectional CCCP predicted by 3D-FEM and 2D-EPM when the concentrated load of 100 N is applied at the center of the plate. It can be seen that the displacements of bidirectional CCCP predicted by 2D-EPM and 3D-FEM under four different boundary conditions are basically the same, and the maximum displacement deviation is only 3.10%. Therefore, the VAM-based equivalent model can be used to evaluate the static behavior of bidirectional CCCP with confidence.

Based on the obtained equivalent stiffness, the buckling critical eigenvalues of the equivalent model obtained by linear buckling analysis are compared with those of 3D-FEM. Table 6 shows that the global buckling modes of 2D-EPM are consistent with those of 3D-FEM, and the buckling critical load error of each mode is less than 4%. Therefore, the VAM-based equivalent model has high accuracy in global buckling analysis of complex convex-concave plates.

To fully verify the correctness of the equivalent model of the bidirectional CCCP, the vibration modes and natural frequencies of 2D-EPM are analyzed by using the SFSF boundary conditions. Table 7 shows the comparison results of the first five vibration modes and natural frequencies between 3D-FEM and 2D-EPM of bidirectional CCCP. It can be seen that the vibration modes of 2D-EPM are consistent with those of 3D-FEM, and the natural frequency error of each mode is less than 2.6%, indicating the VAM-based equivalent model can be confidently used to predict the free vibration of complex convex-concave plates.

It can be concluded from the comparative analysis of Table 1, Table 2, Table 5 and Table 6 that compared with CCCP, the static displacement of bidirectional CCCP decreases and the buckling load increases under the same loading and boundary conditions, indicating the stiffness and stability of the bidirectional CCCP are greatly improved, which is consistent with the results of the previous equivalent stiffness analysis.

## 6. Conclusions

In this work, a VAM-based equivalent model (2D-EPM) is established to predict the effective performance of CCCP. As the approximate energy of 2D-EPM is as close as possible to that of the original 3D plate, it can be used to replace the original CCCP for the global analysis. The following conclusions can be obtained.

(1) The global displacements, buckling modes, and vibration modes of 2D-EPM under different boundary conditions are coincident with those of 3D-FEM, verifying the accuracy of the VAM-based 2D-EPM. Furthermore, the details of local field distribution within unit cell of CCCP are well captured. Compared with CCCP, the static displacement of bidirectional CCCP decreases and the buckling load are significantly increased, indicating the stiffness and stability of the bidirectional CCCP are greatly improved.

(2) The parameter analysis shows that the structural anisotropy of CCCP is more obvious with increasing of the convex height *h*, and the equivalent tensile stiffness decreases gradually, while the change of the equivalent bending stiffness is the opposite of the trend. The equivalent tensile stiffness increases gradually, and the equivalent bending stiffness decreases nonlinearly with increasing of the spacing between adjacent convexities, which may be because the anisotropy of CCCP is not obvious when the spacing between adjacent convexities increases.

(3) The ply angle of composite laminate has a great influence on the equivalent stiffness of CCCP. With increasing of the ply angle, the equivalent tensile and bending stiffness along the $x_{1}$ direction present a nonlinear decrease, while the equivalent tensile and bending stiffness along the $x_{2}$ direction gradually increases, but it has no significant influence on the buckling critical load and natural frequency of CCCP. With the increase in proportion of $0^{\circ }$ ply, the change of tensile and bending stiffness show a trend of linear increase, and the buckling critical load is also increased to a certain extent.

In short, compared with 3D-FEM, the present model has high accuracy for static, buckling, and free vibration analysis of CCCP. At the same time, the DOFs of the 2D-EPM is greatly reduced, resulting in a great improvement in computational efficiency. The present model is particularly useful in the early design stage of composite convex-concave structures where many design trade-offs need to be made over a vast design space in terms of material selection, ply angles, and geometric parameters.

## Figures and Tables

**Figure 1 materials-14-02012-f001:**
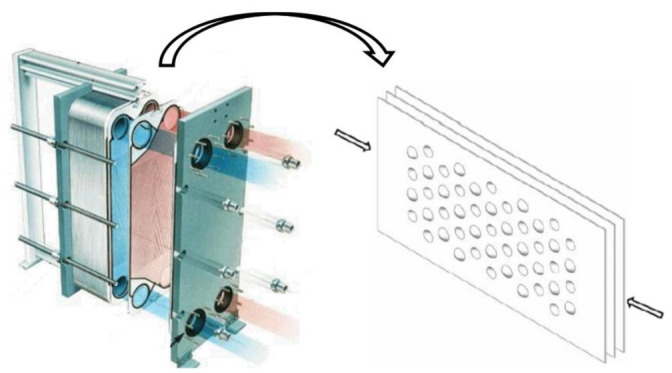
Structural diagram of a concave-convex plate heat exchanger.

**Figure 2 materials-14-02012-f002:**
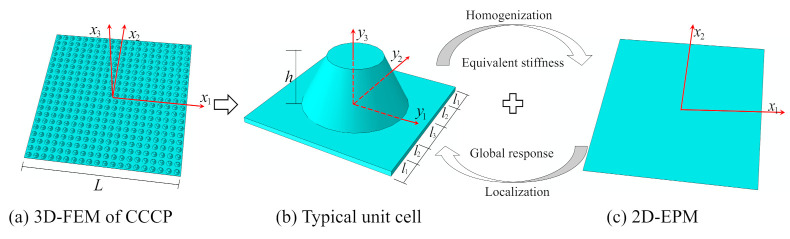
The 3D finite element model (3D-FEM) of the composite conical convex-concave plate (CCCP) (**a**) is divided into a (**b**) 3D unit cell and (**c**) 2D equivalent plate model (2D-EPM).

**Figure 3 materials-14-02012-f003:**
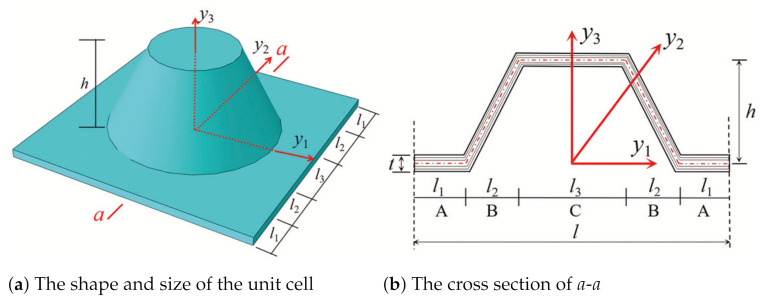
Geometry and sizes of typical unit cell.

**Figure 4 materials-14-02012-f004:**
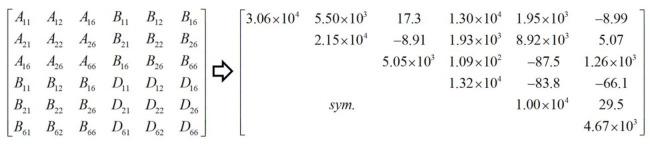
Effective plate properties of CCCP with layup configuration of [0/45/90/−45/0]s obtained by present model.

**Figure 5 materials-14-02012-f005:**
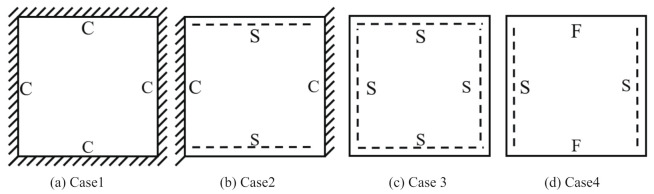
Four boundary conditions used in the static analysis of CCCP: (**a**) CCCC, (**b**) CSCS, (**c**) SSSS, and (**d**) SFSF, with C denoting clamped support, F for free support, and S for simple support.

**Figure 6 materials-14-02012-f006:**
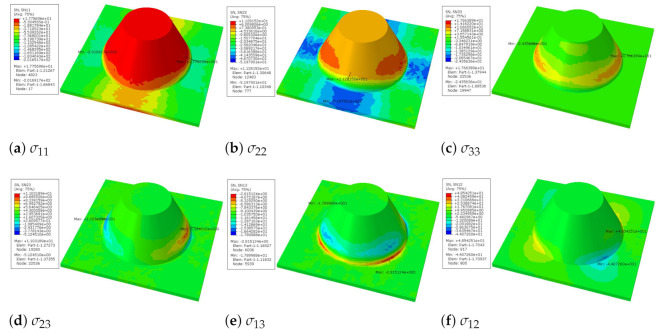
Local stress distributions within the unit cell closest to the center of the plate in Case 1.

**Figure 7 materials-14-02012-f007:**
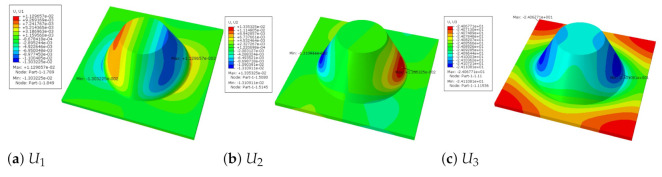
Local displacement distributions within the unit cell closest to the center of the plate in Case 1.

**Figure 8 materials-14-02012-f008:**
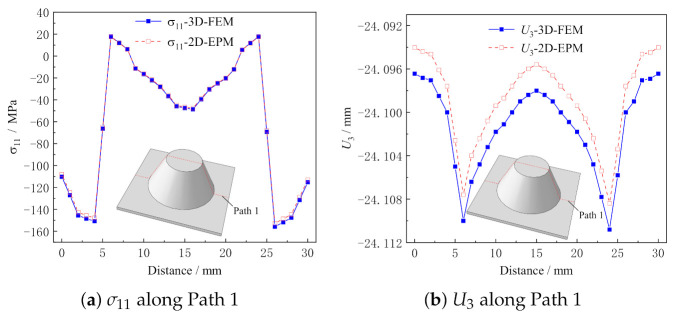
Comparison of $\sigma_{11}$ and $U_{3}$ distribution along Path 1 predicted by 2D-EPM and 3D-FEM.

**Figure 9 materials-14-02012-f009:**
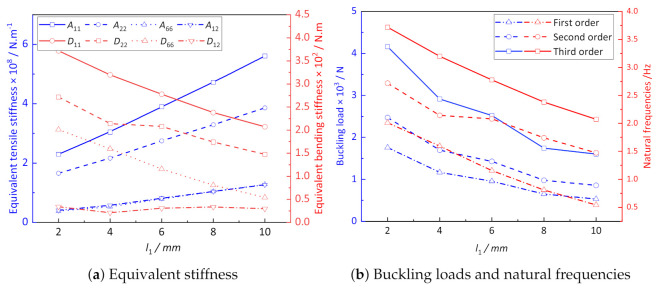
The influence of $l_{1}$ on the effective performance of CCCP.

**Figure 10 materials-14-02012-f010:**
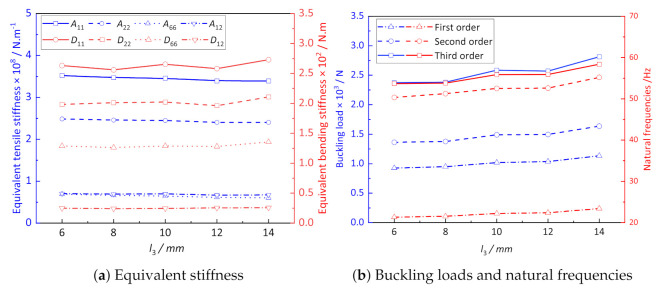
The influence of $l_{3}$ on the effective performance of CCCP.

**Figure 11 materials-14-02012-f011:**
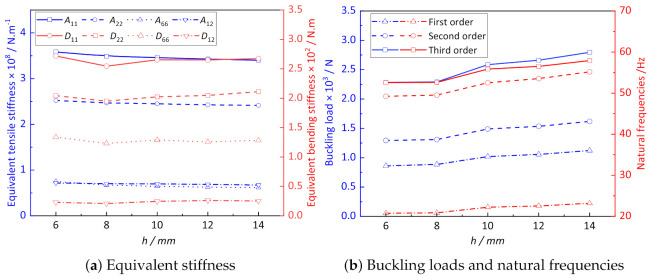
The influence of *h* on the effective performance of CCCP.

**Figure 12 materials-14-02012-f012:**
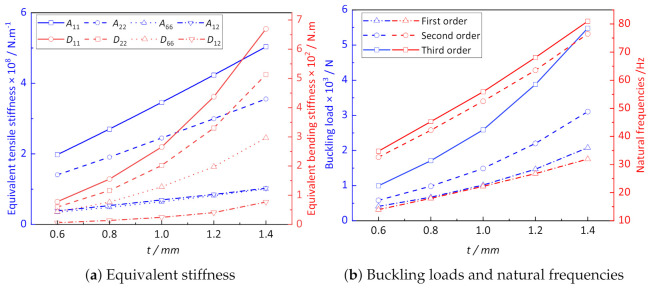
The influence of *t* on the effective performance of CCCP.

**Figure 13 materials-14-02012-f013:**
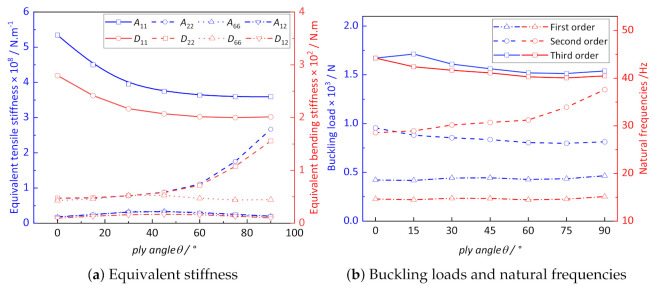
The influence of ply angle on the effective performance of CCCP.

**Figure 14 materials-14-02012-f014:**
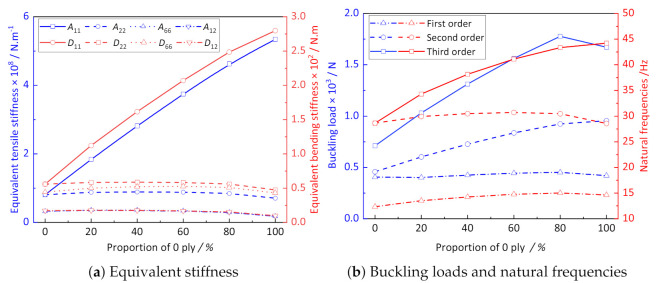
The influence of proportion of $0^{\circ }$ ply on the effective performance of CCCP.

**Figure 15 materials-14-02012-f015:**
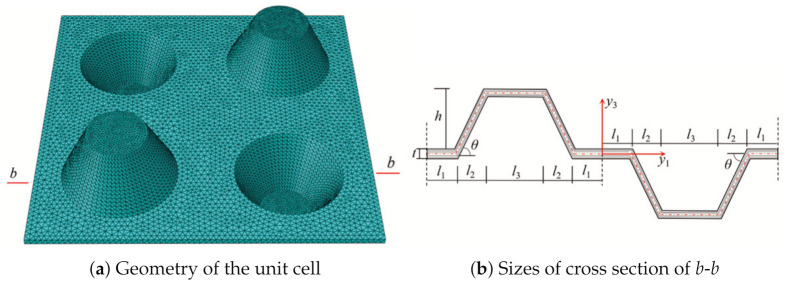
Geometry and sizes of typical unit cell within bidirectional CCCP.

**Figure 16 materials-14-02012-f016:**
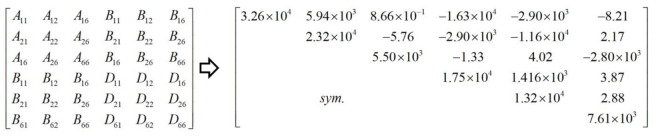
Equivalent stiffness matrix of bidirectional CCCP with layup configuration of [0/45/90/–45/0]s obtained by present model.

**Table 1 materials-14-02012-t001:** Comparison of static displacement between 3D-FEM and 2D-EPM under different boundary conditions (unit: mm).

Cases	Case 1	Case 2	Case 3	Case 4
3D-FEM	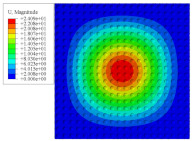	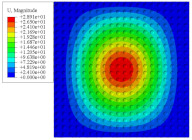	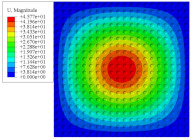	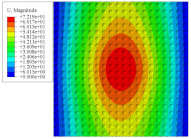
2D-EPM	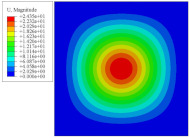	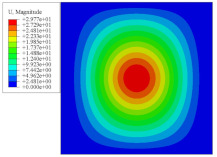	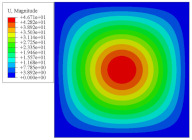	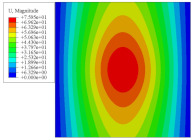
Error %	1.03	4.28	2.05	5.21

**Table 2 materials-14-02012-t002:** Comparison of the first five buckling mode of CCCP in Case 3 predicted by different models (unit: N).

Orders	1	2	3	4	5
2D-EPM	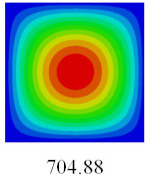	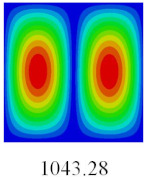	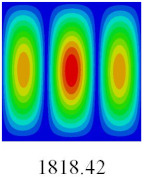	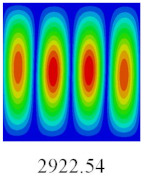	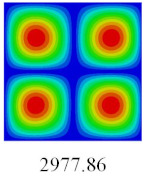
3D-FEM	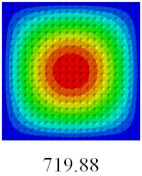	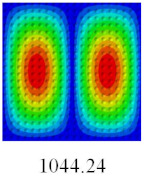	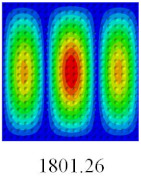	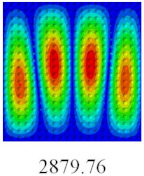	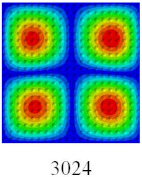
Error %	2.08	0.09	0.95	1.49	1.53

**Table 3 materials-14-02012-t003:** Comparison of natural frequencies (Hz) and mode shapes of CCCP in Case 3 predicted by different models.

Orders	1	2	3	4	5
2D-EPM	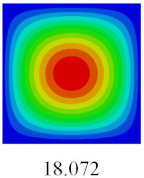	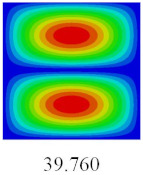	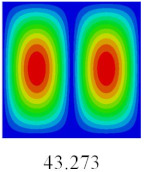	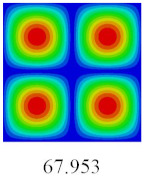	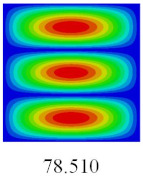
3D-FEM	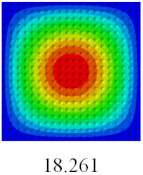	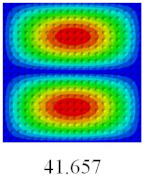	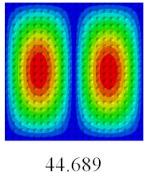	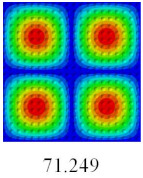	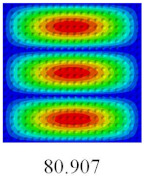
Error %	1.03	4.55	3.17	4.63	2.96

**Table 4 materials-14-02012-t004:** Comparison of computational efficiency between 3D-FEM and 2D-EPM.

Modelling Information		3D FEM	2D-EPM	
			Unit Cell	2D Plate
Element type		C3D10	C3D4	S4R
Number of element		197,356	42,720	14,400
Number of nodes		394,157	59,827	14,641
Running time	Static analysis	5 min 30 s	10 s	30 s
	Buckling analysis	10 min 30 s		25 s
	Free-vibration analysis	7 min 15 s		25 s

**Table 5 materials-14-02012-t005:** Comparison of displacements of bidirectional CCCP under different boundary conditions predicted by 2D-EPM and 3D-FEM.

Cases	Case 1	Case 2	Case 3	Case 4
3D-FEM	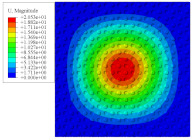	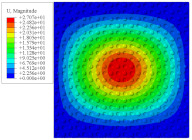	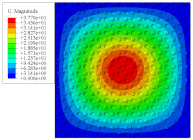	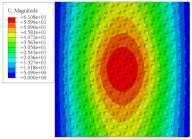
2D-EPM	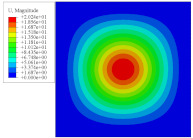	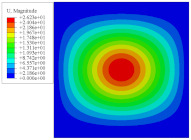	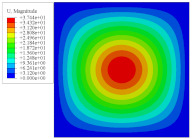	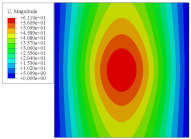
Error %	1.41	3.10	0.69	0.18

**Table 6 materials-14-02012-t006:** Comparison of the first five global buckling modes and critical loads (N) of bidirectional CCCP in Case 3 predicted by different models.

Orders	1	2	3	4	5
2D-EPM	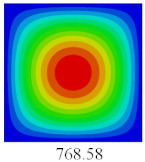	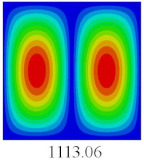	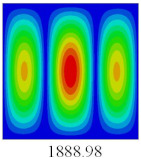	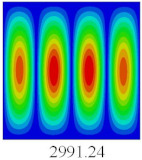	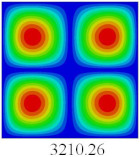
3D-FEM	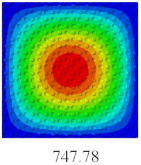	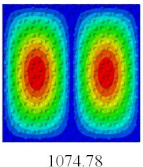	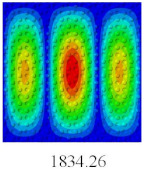	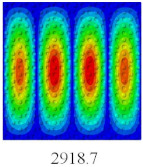	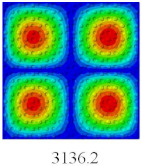
Error %	2.78	3.56	2.98	2.48	2.36

**Table 7 materials-14-02012-t007:** Comparison of the first five vibration modes and natural frequencies (Hz) of bidirectional CCCP under the SFSF boundary conditions predicted by different models.

Orders	1	2	3	4	5
2D-EPM	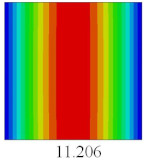	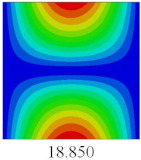		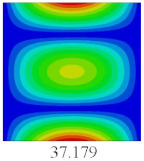	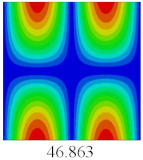
3D-FEM	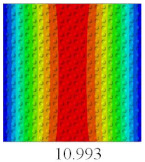	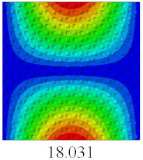		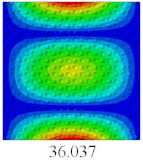	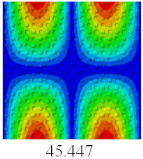
Error %	1.76	2.15	2.66	2.34	2.59

## Data Availability

Data sharing not applicable.

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
