# Peer review of "Structural Analysis of Composite Conical Convex-Concave Plate (CCCP) Using VAM-Based Equivalent Model"

_materials, 2021, doi:10.3390/ma14082012_

Round 1
Reviewer 1 Report
The article „Structural Analysis of Composite Conical Convex-concave Plate (CCCP) using VAM-based Equivalent Model” presents the numerical model of homogenization of anisotropic plates with microstructure, by applying the variational asymptotic method to determine the equivalent stiffness of the unit cell, and transforming the 3D model into 2D equivalent plate model.
The derived algorithm is tested and verified by comparing results (static displacements, buckling loads, and natural frequencies) under different boundary conditions obtained with the proposed algorithm and classical 3D FEM.
General comments
- The analyzed subject-matter falls into the scope of the Materials journal. The presented method of modeling composite plates with conical convex-concave microstructure is novel and interesting. The presented validating examples prove the numerical efficiency and accuracy of the method and its advantage over the classical 3D FEM approach in many applications.
- The presentation of the subject is thorough and correctly designed.
- Due to a large number of introduced symbols and simplifications in mathematical notations, a list of used nomenclature should be introduced.
- Imprecise usage of mathematical symbols and inaccurate equation references (see specific comments) made it difficult to follow the line of reasoning.
- The reference list should be augmented, especially the latest works on the VAM applications should be cited. There are only 2 citations of articles from the last three years.
Specific comments
- Page 4 line 1 - parameter ξ - is not a small parameter, it is a ratio of two scales, (1/ξ is small).
- The symbols are inconsistent in formula 3 the subscript xa appears instead of xα
- In the same formula it should be v3,a instead of v3,2 and the range of a is not defined (or it should be replaced with α )
- In lines 114 and 115, the traction force τi is replaced by the undefined symbol ai
- The undefined symbol <<.>> appears in formula 16 (line 121)
- In formula 16 the term containing Dt is underlined, hence it should be ignored. But in the variational expression (18) the variation from this term is taken into account.
- In formula (24) bars over material matrices should look the same as in (22).
- Line 136. Equation (36) should refer to equation (2) not (4).
- Line under formula (42) should be d2v/dt2.
- Formula (43) should refer to equations (42) and (41)
- In equation (45) should contain va not ve.
- Formula (46) is the simplification of equation (44) not (43).
- In section 3.2 it would be instructive to present the comparison of the local stress distribution recovered from the 2D EPM model with the stresses obtained with the 3D FEM model.
- Is the buckling analysis (section 3.3) determined under the assumption of small displacements or large displacements?
- Section 3.4 – the description of boundary conditions is referred to as defined in the previous section, where 3 different types of boundary conditions were analyzed.
- In section 4.1 the influence of cell geometry parameters L1, L3, and H is discussed separately. If the total length of the unit cell and the conical angle are constant, the change in L1 results in the change of L3 or H (and vice versa). The authors should explain how the influence of the single geometry dimension is interpreted.
Overall recommendation
The paper is suitable for publication after minor revisions.
Reviewer 2 Report
The subject presented in this paper is worthy of investigation, the paper is well written and the conclusions are supported by the data. The reviewer suggests a minor revison in order to correct some typos highlithed in the pdf attached. In particular, the reviewer asks to add a comparison with 3D-FEM solution in Figure 6 and 7.
